# ON THE FLOW MATCHING INTERPRETABILITY

## ABSTRACT

Generative models based on flow matching have demonstrated remarkable success in various domains, yet they suffer from a fundamental limitation: the lack of interpretability in their intermediate generation steps. In fact these models learn to transform noise into data through a series of vector field updates, however the meaning of each step remains opaque. We address this problem by proposing a general framework constraining each flow step to be sampled from a known physical distribution. Flow trajectories are mapped to (and constrained to traverse) the equilibrium states of the simulated physical process. We implement this approach through the 2D Ising model in such a way that flow steps become thermal equilibrium points along a parametric cooling schedule.

Our proposed architecture includes an encoder that maps discrete Ising configurations into a continuous latent space, a flow-matching network that performs temperature-driven diffusion, and a projector that returns to discrete Ising states while preserving physical constraints.

We validate this framework across multiple lattice sizes, showing that it preserves physical fidelity while outperforming Monte Carlo generation in speed as the lattice size increases. In contrast with standard flow matching, each vector field represents a meaningful stepwise transition in the 2D Ising model's latent space. This demonstrates that embedding physical semantics into generative flows transforms opaque neural trajectories into interpretable physical processes.

## 1 INTRODUCTION

Flow matching models have achieved remarkable success in generative tasks, learning to transform noise into complex data distributions through sequential vector field updates (Lipman et al. (2022); Liu et al. (2022)). However, these models suffer from a fundamental interpretability limitation: **the intermediate steps in the generation process lack clear semantic meaning**. While each flow step updates the current state toward the target distribution, what these updates represent remains opaque, limiting our understanding of the learned generative process (Grathwohl et al. (2018)).

This interpretability gap extends beyond flow matching to other generative approaches. Diffusion models (Ho et al. (2020)) and Schrödinger bridges (De Bortoli et al. (2021)) similarly transform data through sequential updates whose individual meaning is unclear. Understanding these intermediate representations is crucial for model debugging, control, and scientific applications where the generation process itself carries important information.

We propose a general framework to address this limitation by constraining flow trajectories to traverse physically meaningful intermediate states. Specifically, we map each flow step to equilibrium states of a known physical process, making the generative trajectory interpretable as a sequence of well-defined physical transitions. This approach transforms the abstract vector field evolution into a concrete, physically-sound process.

We validated this methodology using the 2D Ising model (Ising (1925)), one of the most fundamental systems in statistical mechanics. The Ising model describes interacting boolean spins on a lattice, exhibiting rich temperature-dependent behavior from disordered high-temperature states to ordered low-temperature configurations (Kennett (2020)). During a cooling schedule, the Ising model naturally evolves through successive thermal equilibrium states. Inspired by this, our methodology is designed to mirror this progression, interpreting each diffusive step as an intermediate stage along the cooling path. By mapping flow steps to thermal equilibrium points along a cooling

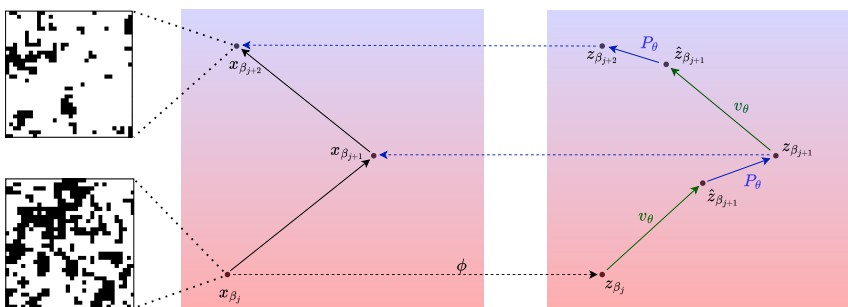

Figure 1: Pipeline architecture showing two consecutive cooling steps. On the left, the spin state, on the right, the latent one. Starting from spin configuration $\boldsymbol{x}^i_{\beta_j}$, the encoder $\phi$ maps it to latent space $\boldsymbol{z}^i_{\beta_j}$, the vector field $v_\theta$ predicts the next latent state $\boldsymbol{z}^i_{\beta_{j+1}}$, and the projector $P_\theta$ decodes back to spin configuration $\boldsymbol{x}^i_{\beta_{j+1}}$. This process is iterated autoregressively through the cooling schedule.

schedule, each vector field update becomes a meaningful thermodynamic transition. Our method is schematically represented in Figure 1.

The key advantage of this approach is that flow trajectories become interpretable, eliminating the black-box nature that limits model trust in domains requiring controllable processes, such as clinical decision-making where each diagnostic step must be traceable for patient safety (Ennab & Mcheick (2024); Bereska & Gavves (2024)). With our approach each step corresponds to a specific temperature transition in the Ising model's phase space. This interpretability extends beyond the specific physical system, providing a general methodology for constraining generative flows to meaningful intermediate representations.

The contributions of this work are as follows: *(i)* We propose a framework that constrains flow matching models to traverse equilibrium states of a physical process, giving semantic meaning to each generative step, *(ii)* we instantiate this idea with the 2D Ising model, where flow updates correspond to thermal transitions along a cooling schedule, and *(iii)* we show that our method preserves physical fidelity while scaling more efficiently than Monte Carlo sampling, and turns otherwise opaque flows into interpretable physical trajectories.

We evaluated our approach theoretically and empirically, demonstrating preservation of physical fidelity while achieving computational efficiency superior to traditional Monte Carlo methods.

## 2 RELATED WORKS

Flow matching (Lipman et al. (2022); Liu et al. (2022)) has recently emerged as a powerful alternative to diffusion models (Ho et al. (2020); Asperti et al. (2023)) by learning continuous deterministic trajectories through optimal transport. This framework has been extended to physical systems (Baldan et al. (2025)), discrete spaces (Gat et al. (2024)), and discrete lattice models (Tuo et al. (2025)). However, these models suffer from a fundamental interpretability limitation: the intermediate steps in the generation process lack clear semantic meaning (Grathwohl et al. (2018)).

The interpretability challenge in generative models has attracted significant attention through disentanglement studies (Higgins et al. (2017); Locatello et al. (2019)), interpretable latent interpolations (Chen et al. (2016); Karras et al. (2019)), and for flow-based models, visualization of learned transformations (Dinh et al. (2017)) and manifold geometry analysis (Kumar & Poole (2020)). Recent approaches employ physics-informed constraints (Greydanus et al. (2019)) and symmetry-preserving architectures (Köhler et al. (2020)) for enhanced interpretability, yet intermediate flow steps remain abstract without clear physical correspondence.

Generating configurations for models like Ising has historically relied on Monte Carlo methods, from classical Metropolis-Hastings (Metropolis et al. (1953); Hastings (1970)) to cluster algorithms like Swendsen-Wang (Swendsen & Wang (1987)) and Wolff (Wolff (1989)) that partially address critical slowing down. Machine learning approaches have enhanced these through improved

reconstruction (Matthews et al. (2022)) and accelerated sampling (Levy et al. (2021); Nicoli et al. (2021)), though still requiring traditional equilibration procedures.

The intersection of neural networks and statistical mechanics originates with Hopfield networks (Hopfield (1982)), where spin-glass models informed associative memory, with subsequent work employing neural architectures to study physical systems and phase transitions (Mézard et al. (1987); Amit et al. (1985)). Supervised learning has been applied to classify phases of matter (Carrasquilla & Melko (2017); Ponte & Melko (2017)), while variational approaches using Restricted Boltzmann Machines enabled quantum many-body state representation (Carleo & Troyer (2017); Torlai et al. (2018)). Variational autoencoders (Kingma et al. (2013); Asperti et al. (2021)) have discovered low-dimensional latent representations of physical systems (Wetzel (2017); Wang (2016)), and generative adversarial networks (Goodfellow et al. (2020)) have accelerated MC dynamics (Nicoli et al. (2020)). Autoregressive models directly estimate Boltzmann distributions (Wu et al. (2019); Hibat-Allah et al. (2020)), while flow-based models demonstrate improved sampling efficiency in lattice field theories (Albergo et al. (2019); Boyda et al. (2021)). Notably, D'Angelo & Böttcher (2020) proposed neural generative models for sampling Ising configurations at fixed temperatures, though without guaranteeing consistency in sequential generation across temperature decreases.

To the best of our knowledge, we are the first to address the interpretability gap in flow matching by constraining each flow step to correspond to meaningful physical transitions. We used the Ising model's thermal evolution as an example to validate our idea. This transforms abstract vector field evolution into physically meaningful processes where intermediate steps represent well-defined equilibrium states along temperature trajectories.

## 3 Preliminaries

In this section, we first introduce the notation used throughout the paper. Then, we will briefly recall the main topic relevant to this work.

**Notations** To simplify the discussion, we adopt the following notations. Let $\boldsymbol{x} \in \{-1, +1\}^{N \times N}$ denote a binary two-dimensional grid. We represent its elements in lexicographic order as $x_i$, for $i = 1, \ldots, N^2$, where $x_i$ refers to the $i$-th entry of $\boldsymbol{x}$.

Given a function $f : \{-1, +1\}^{N \times N} \to \mathbb{R}$, we denote by $\langle f(\boldsymbol{x}) \rangle$ the empirical expectation of $f(\boldsymbol{x})$, computed by averaging its values over a number of independent samples of $\boldsymbol{x}$.

Finally, for any index $i = 1, \ldots, N^2$, we define $U_i$ as the set of neighboring indices of $i$, i.e., $U_i = \{j \in \{1, \ldots, N^2\} \mid d(i, j) = 1\}$, where $d(i, j)$ denotes the Manhattan distance in two dimensions (Deza & Deza (2009)).

### 3.1 Flow Matching

Flow Matching (FM) (Lipman et al. (2022)) represents a deterministic generalization of diffusion models, where the evolution of samples over time is governed not by a stochastic process, but by a learned time-dependent vector field. While in diffusion models the forward process progressively corrupts data into noise via a stochastic differential equation (SDE) (Song et al. (2020)) such as

$$d\boldsymbol{x}_t = f(\boldsymbol{x}_t, t)dt + g(t)d\boldsymbol{w}_t, \tag{1}$$

and generation is achieved by reversing this process, typically requiring score estimation and iterative sampling, Flow Matching works by defining a continuous trajectory $\boldsymbol{x}_t := \kappa_t(\boldsymbol{x}_0)$, mapping a sample $\boldsymbol{x}_0 \sim p_0(\boldsymbol{x}_0)$ from a simple base distribution (e.g. $p_0(\boldsymbol{x}_0) = \mathcal{N}(0, I)$) toward the data distribution $\boldsymbol{x}_1 \sim p_1(\boldsymbol{x}_1)$, and seeks to learn a neural vector field $v_\theta(\boldsymbol{x}, t)$ that drives this transformation deterministically via an ordinary differential equation (ODE):

$$d\boldsymbol{x}_t = u(\boldsymbol{x}_t, t)dt. \tag{2}$$

Then, Flow Matching trains $v_\theta$ through the regression loss:

$$\mathcal{L}_{\text{FM}}(\theta) = \mathbb{E}_{t \sim \mathcal{U}[0,1], \, \boldsymbol{x}_t \sim p_t} \left[ \| v_\theta(\boldsymbol{x}_t, t) - u(\boldsymbol{x}_t, t) \|^2 \right],$$

where $u(\boldsymbol{x}_t, t) = \frac{d\boldsymbol{x}_t}{dt}$ from Equation 2, and $p_t$ is implicitly defined by the choices of $\kappa_t$ and $p_0$.

A particularly efficient and stable variant, corresponding to the optimal transport (OT) mapping between source and target distribution, arises when the path $\kappa_t$ is chosen as the linear interpolation between $\boldsymbol{x}_0 \sim p_0(\boldsymbol{x}_0)$ and $\boldsymbol{x}_1 \sim p_1(\boldsymbol{x}_1)$, i.e., $\boldsymbol{x}_t := \kappa_t(\boldsymbol{x}_0) = (1 - t)\boldsymbol{x}_0 + t\boldsymbol{x}_1$. In this case, the trajectory follows a straight line in data space, and the corresponding velocity field $u(\boldsymbol{x}_t, t)$ becomes constant, as $u(\boldsymbol{x}_t, t) = \frac{d\boldsymbol{x}_t}{dt} = \boldsymbol{x}_1 - \boldsymbol{x}_0$, allowing it to be known analytically and used directly as supervision, avoiding simulation of the dynamics during training.

While the choice of $\kappa_t(\boldsymbol{x}_0)$ as a linear interpolation between the source and target distribution is well suited for general flow matching applications such as unconditional image generation, it cannot be used directly in our setup as we require $\boldsymbol{x}_t$ to be an admissible Ising configuration at given prescribed timestep $t \in \{t_0, \ldots, t_D\}$ where $D$ is the fixed number of diffusion steps. For this reason, in this work we define $\kappa_t(\boldsymbol{x}_0)$ to be a piecewise linear interpolation between Ising configurations $\boldsymbol{x}_{t_j}$, i.e.

$$\kappa_t(\boldsymbol{x}_0) = \frac{t_{j+1} - t}{t_{j+1} - t_j}\boldsymbol{x}_{t_j} + \frac{t - t_j}{t_{j+1} - t_j}\boldsymbol{x}_{t_{j+1}} \tag{3}$$

for any $t \in [t_j, t_{i+j}]$, where $\boldsymbol{x}_{t_j}$ represents the state of the Ising model at temperature $t_j$. With this choice of $\kappa_t(\boldsymbol{x}_0)$, $u(\boldsymbol{x}_t, t) = \frac{\boldsymbol{x}_{t_{j+1}} - \boldsymbol{x}_{t_j}}{t_{j+1} - t_j}$, which preserves the property of being known analytically as with the OT mapping, while allowing the dynamics to pass through the prescribed trajectory.

## 3.2 THE ISING MODEL

The squared Ising model is composed of a grid with dimensions $N \times N$, where each cell $i$ in the grid has a spin $\sigma_i \in \{-1, +1\}$. This two-dimensional square lattice system is characterized by the Hamiltonian:

$$H(\boldsymbol{x}) = -\frac{J}{2}\sum_{i=1}^{N^2}\sum_{j \in U_i} x_i x_j - h\sum_{i=1}^{N^2} x_i, \tag{4}$$

where $J > 0$ is the ferromagnetic coupling constant, and $h$ denotes an external magnetic field.

The different configurations of the grid depend not only on the Hamiltonian itself but, from a statistical mechanics perspective, also on an additional parameter, the temperature $T$, which represents the thermal bath in which the system is placed.

We introduce the inverse temperature parameter $\beta = 1/(k_B T)$ to incorporate the effect of temperature. Under this formulation, the probability of observing a given configuration $\sigma$ is governed by the Gibbs probability at inverse temperature $\beta$, defined as $\mathbb{P}_\beta(\boldsymbol{x}) = \frac{1}{Z(\beta)}e^{-\beta H(\boldsymbol{x})}$, where $Z(\beta)$ is the partition function, ensuring normalization of the probability distribution.

The system displays disordered configurations in the grid at small values of $\beta$ (i.e., high temperatures), indicative of a paramagnetic regime (Hasenbusch et al. (2007)). Large values of $\beta$ (low temperatures), in contrast, lead to ordered configurations, characteristic of ferromagnetic behavior. As the temperature $T$ reaches critical temperature $T_c \approx 2.269$, the system undergoes a phase transition marked by abrupt changes in observable quantities. This correspond to the emergence of clusters of aligned spins on the grid.

The critical temperature $T_c$ is theoretically calculated via $\beta_c$, defined as $\beta_c := \frac{1}{k_B T_c}$, whose value can be obtained by the Onsager's solution (Onsager (1944)) $\beta_c = \frac{1}{2J}\ln(1 + \sqrt{2})$. In the Appendix we include the Onsager exact derivation. In this paper we consider the typical setup where the external field is set to zero ($h = 0$) to preserve the spin-flip symmetry of the model, allowing investigation of spontaneous symmetry breaking and intrinsic collective behavior.

Because the configuration space grows exponentially with system size ($2^N$), exact analytical solutions or enumerations become intractable. Therefore, Monte Carlo methods are employed to efficiently sample from $\mathbb{P}_\beta(\boldsymbol{x})$, enabling numerical estimation of macroscopic observables such as magnetization and correlation functions, especially near the critical temperature.

### 3.3 MONTE CARLO METHODS

We employ two well-established Monte Carlo algorithms to generate the dataset used in this study: the Metropolis (Metropolis et al. (1953)), and the Wolff cluster algorithm (Wolff (1989)). The Metropolis algorithm is one of the most widely used approaches due to its simplicity and general applicability in simulating spin systems. However, it is known to suffer from critical slowing down near phase transitions as the temperature decreases. The Wolff algorithm was developed to address some of these limitations by efficiently updating clusters of spins rather than individual ones.

The Metropolis method proceeds by proposing single-spin flips, which are accepted or rejected based on the associated change in the system's energy. Given two grids, we define the energy change between the two as $\Delta E_i = 2x_i \sum_{\langle i,j \rangle} x_j$. If $\Delta E_i \leq 0$, the flip is accepted unconditionally; otherwise, it is accepted with probability $p_i = e^{-\beta \Delta E_i}$. This procedure is repeated for all sites in a randomized order, ensuring fair sampling of the configuration space. Periodic boundary conditions are applied throughout.

In contrast, the Wolff algorithm constructs and flips entire clusters of aligned spins at once, significantly reducing times and allowing better escape from local minima. By flipping clusters instead of single spins, the algorithm overcomes some of the slow dynamics encountered by Metropolis. It begins by selecting a random lattice cell $i$. A cluster is initialized containing this cell, and a stack is created to manage the cluster growth. While the stack is not empty, a site $j$ is removed from it, and each of its nearest neighbors $k \in U_j$ is examined. If $x_k = x_j$ and $k$ is not already in the cluster, then $k$ is added to the cluster with probability $p_k = 1 - e^{-2\beta}$. When a neighbor is added, it is also pushed onto the stack to continue cluster growth. After the cluster is fully constructed, all spins in it are flipped simultaneously. Periodic boundary conditions are applied throughout for both algorithms.

## 4 METHODOLOGY

### 4.1 DATASET CONSTRUCTION

We construct a dataset capturing spin configurations across thermal conditions, including the cooling dynamics of the two-dimensional Ising grid around the critical point $T_c$. Our system is defined by Eq. (4), assuming an external magnetic field $h = 0$ in accordance with Onsager's solution. Without loss of generality, and up to a multiplicative scaling factor, we set the coupling constant $J = 1$.

Square lattices with linear sizes $N \in \{32, 48, 64\}$ were used to capture critical phenomena across multiple scales. The dataset is generated via simulated annealing from $T_{\max} = 5$ down to $T_{\min} = 1$ over $D = 20$ linearly spaced temperatures. The chosen temperature range spans the ordered ferromagnetic phase ($T < 2.0$), the critical region ($T_c \approx 2.269$), and the disordered paramagnetic phase ($T > 2.5$).

Sampling employs the Wolff algorithm subjected to a magnetization sign constraint to prevent unlikely sign changes (in nature). At each temperature, the system is equilibrated until convergence, defined by the relative error criterion $\frac{|H_{\text{sim}}(\boldsymbol{x}) - H_{\text{exact}}|}{|H_{\text{exact}}|} < \epsilon$, where $\epsilon = 0.05$, $E_{\text{sim}}$ is the simulated energy per spin and $E_{\text{exact}}$ is Onsager's exact solution given by:

$$H_{\text{exact}} = -\frac{\cosh(2\beta)}{\sinh(2\beta)} \left[ 1 + \frac{2}{\pi} \left( 2 \tanh^2(2\beta) - 1 \right) I(k) \right],$$

with $k = 2\sinh(2\beta)/\cosh^2(2\beta)$ and $I(k)$ the complete elliptic integral of the first kind. We compute classic thermodynamic observables in order to assess dataset fidelity:

$$E = \frac{\langle H(\boldsymbol{x}) \rangle}{L^2}, \quad m = \frac{\langle |M(\boldsymbol{x})| \rangle}{L^2}, \quad C_v = \frac{\beta^2 \langle (\Delta H(\boldsymbol{x}))^2 \rangle}{L^2}, \quad \chi = \frac{\beta \langle (\Delta M(\boldsymbol{x}))^2 \rangle}{L^2}, \quad (5)$$

where $M = \sum_i \sigma_i$ is the magnetization. The estimated critical temperature is $T_c^{\text{est}} = 2.31 \pm 0.05$, in agreement with the exact Onsager result $T_c \approx 2.269$ when finite-size effects are considered.

Let $\{\beta_j\}_{j=0}^D$ be the set of inverse temperatures corresponding to a cooling schedule from $T_{\max}$ to $T_{\min}$. We define a *trajectory* as a sequence of Ising spin configurations sampled at these inverse temperatures $\Gamma^i = \left\{ \boldsymbol{x}_{\beta_0}^i, \boldsymbol{x}_{\beta_1}^i, \ldots, \boldsymbol{x}_{\beta_D}^i \right\}$, where $\boldsymbol{x}_{\beta_j}^i$ denotes the configuration of trajectory $i$ at inverse temperature $\beta_j$. For each transition $(\boldsymbol{x}_{\beta_j}^i, \boldsymbol{x}_{\beta_{j+1}}^i)$, we additionally store $K = 40$ independent cooling Monte Carlo samples to estimate the conditional distribution of each lattice site.

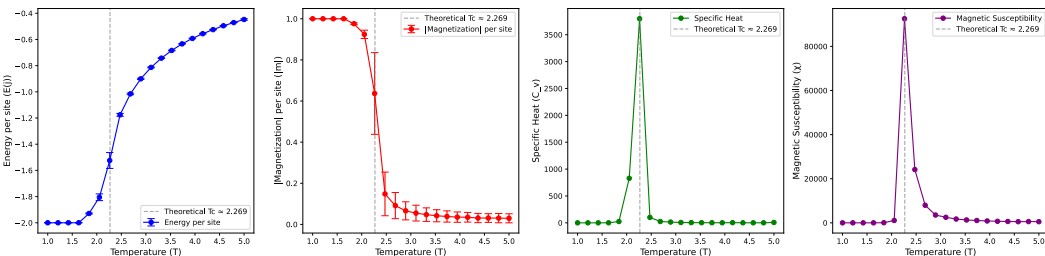

Figure 2: Thermodynamic observables for the $N = 48$ lattice as a function of temperature. The vertical dotted line indicates the critical temperature.

## 4.2 PROPOSED METHOD

As already anticipated, our objective is to transition from a high-temperature distribution to a low-temperature one via a piecewise linear flow matching model which traverses equilibrium states of the Ising model at prescribed inverse temperatures $\{\beta_0, \beta_1, \ldots, \beta_D\}$.

Since flow matching models are designed to work on continuous spaces, we begin by embedding the discrete Ising grids $\boldsymbol{x}_{\beta_j}^i$ into continuous latent representations $\boldsymbol{z}_{\beta_j}^i = \phi(\boldsymbol{x}_{\beta_j}^i)$ via a learned function $\phi$, whose detail will be given below.

The problem thus reduces to learning the slope between successive cooling steps in the latent space $\boldsymbol{z}_{\beta_j}^i$ via a neural network $v_\theta(\boldsymbol{z}_{\beta_j}^i, \beta_j)$ satisfying:

$$d\boldsymbol{z}_{\beta_j}^i = v_\theta(\boldsymbol{z}_{\beta_j}^i, \beta_j)d\beta_j.$$

As discussed in Equation 3, the piecewise linear choice for the dynamics $\boldsymbol{z}_{\beta_j}^i := \kappa_{\beta_j}(\boldsymbol{z}_{\beta_0}^i)$ implies that:

$$v_\theta\left(\boldsymbol{z}_{\beta_j}^i, \beta_j\right) \approx \frac{\boldsymbol{z}_{\beta_{j+1}}^i - \boldsymbol{z}_{\beta_j}^i}{\beta_{j+1} - \beta_j} = \frac{\phi\left(\boldsymbol{x}_{\beta_{j+1}}^i\right) - \phi\left(\boldsymbol{x}_{\beta_j}^i\right)}{\beta_{j+1} - \beta_j}. \qquad (6)$$

Each vector field update $v_\theta(z_{\beta_j}, \beta_j)$ thus represents a physically meaningful thermal transition, transforming opaque latent dynamics into interpretable thermodynamic processes. Since multiple Monte Carlo trajectories may produce different cooled states from the same initial configuration, we characterize the cooling dynamics in the latent space by computing an expected forward transition. Specifically, for a given embedded configuration $\boldsymbol{z}_{\beta_j}^i$, we define the expected target vector field as the average $K$ such transitions:

$$\left\langle v\left(\boldsymbol{z}_{\beta_j}^i, \beta_j\right)\right\rangle := \frac{1}{K} \sum_{k=1}^K \frac{\boldsymbol{z}_{\beta_{j+1}}^{i,k} - \boldsymbol{z}_{\beta_j}^i}{\beta_{j+1} - \beta_j},$$

where $\boldsymbol{z}_{\beta_{j+1}}^{i,k}$ denotes the $k$-th Monte Carlo sample obtained by cooling the configuration $\boldsymbol{z}_{\beta_j}^i$ to the next $\beta_{j+1}$. After training, $v_\theta\left(\boldsymbol{z}_{\beta_j}, \beta_j\right)$ is employed to perform steps between successive cooler grids in the latent space by:

$$\boldsymbol{z}_{\beta_{j+1}}^i = \boldsymbol{z}_{\beta_j}^i + \Delta\beta_j \cdot v_\theta\left(\boldsymbol{z}_{\beta_j}^i, \beta_j\right), \quad \Delta\beta_j := \beta_{j+1} - \beta_j.$$

Note that the values $\boldsymbol{z}_{\beta_{j+1}}^i$ obtained by the equation above does not necessarily corresponds to the latent encoding of an Ising equilibrium grid at inverse temperature $\beta_{j+1}$, as small approximations

error in $v_\theta\left(z_{\beta_j}^i, \beta_j\right)$ may cause some deviation in the prediction of $z_{\beta_{j+1}}^i$. To avoid any amplification of error caused by the iterative procedure, we consider a projection mapping $P_\theta$, representing either a few step of Monte Carlo simulation or a learned decoder, whose aim is to ensure physical plausibility of the generated trajectory and at the same time mapping the latent vector $z_{\beta_{j+1}}^i$ back to the discrete domain, i.e.:

$$\hat{\boldsymbol{x}}_{\beta_{j+1}}^i = P_\theta\left(\boldsymbol{z}_{\beta_{j+1}}^i, \beta_{j+1}\right).$$

We train the encoder $\phi$, the flow matching model $v_\theta$, and the decoder $P_\theta$ separately. In particular, the latent encoder $\phi$ is trained in an autoencoder-like flavor as:

$$\phi, \gamma = \arg\min_{\phi, \gamma} \frac{1}{N_{data}} \sum_{i=1}^{N_{data}} \left\|\gamma\left(\phi\left(\boldsymbol{x}_{\beta_j}^i\right)\right) - \boldsymbol{x}_{\beta_j}^i\right\|_2^2,$$

where the inverse-map $\gamma$ is discarded after training.

In order to train the flow matching model $v_\theta$, we represent the distribution of target vectors $\langle v \rangle$ at each $\left(\boldsymbol{z}_{\beta_j}^i, \beta_j\right)$ point using a kernel density estimator (KDE) as $\mathbb{P}\left(v \mid \boldsymbol{z}_{\beta_j}^i, \beta_j\right) = \mathcal{N}\left(v; \langle v \rangle, \sigma^2 I\right)$, where $\sigma$ controls the kernel bandwidth. The corresponding loss to minimize is the negative log-likelihood under this KDE, i.e.:

$$\mathcal{L}_{\text{KDE}} = \frac{1}{2\sigma^2} \left\|v_\theta\left(\boldsymbol{z}_{\beta_j}^i, \beta_j\right) - \left\langle v\left(\boldsymbol{z}_{\beta_j}^i, \beta_j\right)\right\rangle\right\|^2.$$

Finally, the projection model $P_\theta$ is trained to minimize the MSE loss between grids $\hat{\boldsymbol{x}}_{\beta_j}^i$ and $\boldsymbol{x}_{\beta_j}^i$, plus a regularization term $\mathcal{L}_{\text{Ising}}$ which is added to enforce thermodynamic consistency and respect the Ising model's Hamiltonian, defined as:

$$\mathcal{L}_{\text{Ising}} = \left| H\left(\hat{\boldsymbol{x}}_{\beta_{j+1}}\right) - H\left(\boldsymbol{x}_{\beta_{j+1}}\right)\right|.$$

Predictions proceed autoregressively at test time: we predict $\boldsymbol{x}_{\beta_{j+2}}^i$, the output, from $\boldsymbol{x}_{\beta_j}^i$ through $\boldsymbol{x}_{\beta_{j+1}}^i$. This process is iterated using the full pipeline until the minimum $\beta_0$ is reached.

## 5 EXPERIMENTS

The goal of the experiments we performed is to assess how well the autoregressive predictions generated by the pipeline approximates the ground-truth (GT). Each pipeline consists of a fixed encoder $\phi$ and a fixed latent vector field $v_\theta$, while differing in the decoding mechanism. We evaluate two decoder variants, namely Metropolis-Hastings with (i) 10 (MH-10) and (ii) 15 (MH-15) refinement steps and (iii) the learned decoder ($P_\theta$).

In order to prove that the vector field $v_\theta$ is non-trivial (i.e., not simply the identity function in latent space) and that the chosen decoder does not override or compensate for the contribution of $v_\theta$, we also report a comparison against an autoregressive Metropolis Hasting generation with the same number of refinement steps (15) as the reference decoder (called MC-15). The resulting modular design allows for a plug-and-play architecture, enabling flexible substitution of decoding components and supporting future extensions of our method.

Experiments were conducted on Ising grids of size $N \in \{32, 48, 64\}$. Evaluation includes both qualitative analysis (through visual inspection of spin configurations) and quantitative assessment based on statistical measures of the macroscopic observables of the system.

### 5.1 QUALITATIVE EVALUATION OF THE GRIDS

As shown in Figure 3, we compare spin configurations across temperatures for two decoding approaches: MH-$\{10, 15\}$ and the learned decoder $P_\theta$. Both models reproduce the expected increase in disorder with temperature. However, the MH-based decoder produces more scattered spin patterns that lack coherent structure in the phase transition and fail to capture the cluster formation, even though the resulting samples align with macroscopic observables. In contrast, the learned decoder generates configurations that more closely resembles the true physical behavior, particularly near the critical temperature, where clear spin clusters emerge.

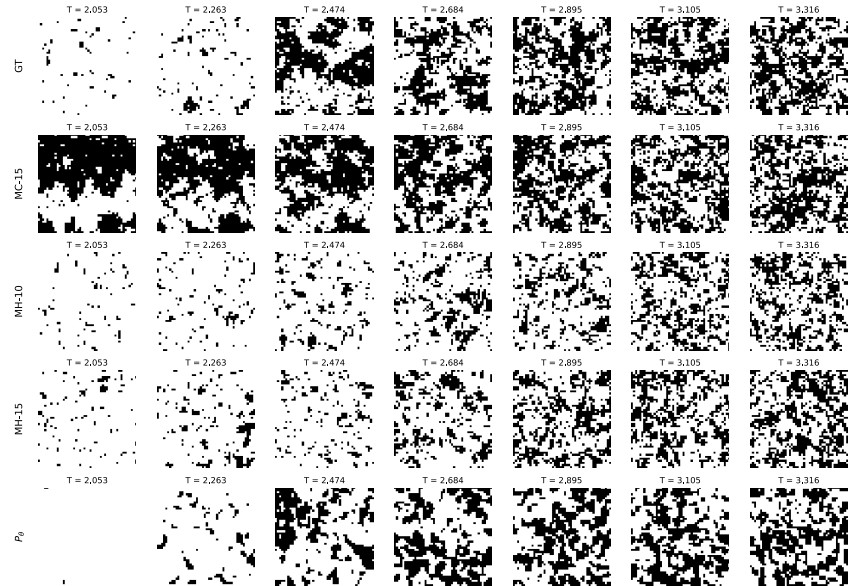

Figure 3: Comparison of decoding methods in the encoder-vector field pipeline across increasing temperature values (left to right) for grid size $N = 48$. Further comparisons for grid sizes $N = 32$ and $N = 64$ are provided in the Technical Appendix.

## 5.2 QUANTITATIVE EVALUATION

We quantitatively evaluate model accuracy by comparing predicted observables $\hat{\boldsymbol{x}}^i_{\beta_j}$ against ground truth values $\boldsymbol{x}^i_{\beta_j}$ from Metropolis Hastings simulations across four thermodynamic quantities described in Eq. 5: energy ($E$), magnetization ($m$), specific heat ($C_v$), and magnetic susceptibility ($\chi$).

| Lattice Size | Method | $\Delta\langle E\rangle(J)$ | $\Delta\langle M\rangle$ | $\Delta\langle C_v\rangle(k_B)$ | $\Delta\langle\chi\rangle(J^{-1})$ | Time (s) |
|---|---|---|---|---|---|---|
| | GT | – | – | – | – | $\approx 7$ |
| | MC-15 | $0.107 \pm 0.120$ | $-0.169 \pm 0.259$ | $0.682 \pm 0.575$ | $8.30 \pm 15.99$ | $0.644 \pm 0.044$ |
| $32 \times 32$ | MH-10 | $0.0252 \pm 0.0365$ | $0.0327 \pm 0.0800$ | $0.885 \pm 1.113$ | $4.179 \pm 7.299$ | $0.540 \pm 0.038$ |
| | MH-15 | $0.0338 \pm 0.0284$ | $0.0210 \pm 0.0644$ | $0.744 \pm 0.912$ | $3.057 \pm 5.519$ | $0.750 \pm 0.0531$ |
| | $P_\theta$ | $0.0214 \pm 0.035$ | $0.068 \pm 0.0928$ | $0.396 \pm 0.293$ | $-0.706 \pm 3.312$ | $0.581 \pm 0.049$ |
| | GT | – | – | – | – | $\approx 20$ |
| | MC-15 | $0.117 \pm 0.126$ | $0.210 \pm 0.313$ | $0.527 \pm 0.570$ | $6.8 \pm 20.3$ | $1.38 \pm 0.0282$ |
| $48 \times 48$ | MH-10 | $-0.0133 \pm 0.0771$ | $0.171 \pm 0.194$ | $0.348 \pm 0.382$ | $-3.44 \pm 10.63$ | $1.00 \pm 0.01$ |
| | MH-15 | $0.0109 \pm 0.0612$ | $0.119 \pm 0.169$ | $0.332 \pm 0.359$ | $-3.40 \pm 10.55$ | $1.48 \pm 0.03$ |
| | $P_\theta$ | $0.00461 \pm 0.00362$ | $0.0395 \pm 0.0567$ | $0.0537 \pm 0.2690$ | $-3.90 \pm 10.27$ | $0.529 \pm 0.034$ |
| | GT | – | – | – | – | $\approx 35$ |
| | MC-15 | $0.113 \pm 0.119$ | $0.231 \pm 0.345$ | $0.609 \pm 0.530$ | $7.30 \pm 18.76$ | $2.27 \pm 0.02$ |
| $64 \times 64$ | MH-10 | $0.123 \pm 0.135$ | $-0.235 \pm 0.351$ | $0.830 \pm 0.744$ | $-1.10 \pm 6.71$ | $1.69 \pm 0.09$ |
| | MH-15 | $0.112 \pm 0.121$ | $-0.232 \pm 0.351$ | $0.593 \pm 0.563$ | $0.234 \pm 7.850$ | $2.57 \pm 0.12$ |
| | $P_\theta$ | $0.0357 \pm 0.0371$ | $0.0088 \pm 0.0437$ | $0.137 \pm 0.149$ | $-2.84 \pm 6.77$ | $0.563 \pm 0.071$ |

Table 1: Performance comparison of sampling methods across different lattice sizes. Reference methods (GT, MC with 15 steps) are compared proposed methods (MH with 10 and 15 steps and $P_\theta$). Mean Absolute Error (%) between measured physical observables and ground truth. The time represents the mean time needed to generate a single cooling trajectory.

As shown in the quantitative results in Table 1, the learned decoder consistently achieves the fastest inference times across all configurations. More importantly, the computational advantage of our learned approach increases with grid size, significantly outperforming Metropolis Hastings methods for larger lattices while maintaining equivalent convergence quality. All developed models demonstrate strong adherence to ground truth values (Figure 4).

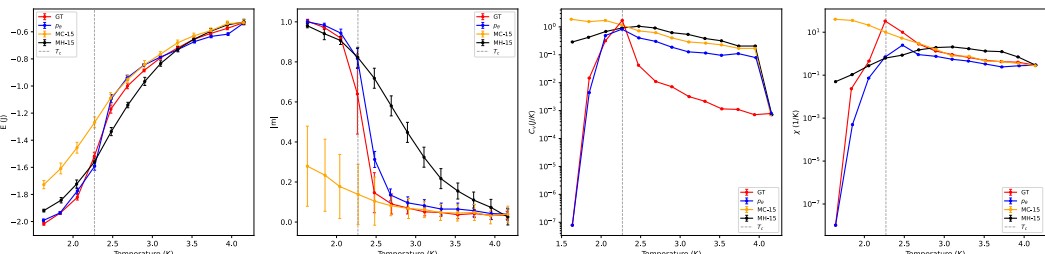

Figure 4: Comparison of physical observables distributions defined in 5 over the cooling schedule (grid size $N = 48$). $C_v$ and $\chi$ on a log scale. The observables for grid size $N = 32$ and $N = 64$ are provided in the Technical Appendix.

The autoregressive MC-15 approach fails to predict low-energy configurations, suggesting some limitations of purely statistical sampling methods. Moreover, its performance degrades as the grid size increases due to the fixed number of sampling steps, which becomes insufficient to explore the exponentially growing configuration space. In particular for $64 \times 64$ grids, keeping the number of steps low to maintain comparable runtimes leads to poor results for all Monte Carlo methods across all physical observables. In contrast, our learned approaches show consistent results with ground truth across all temperature ranges for both energy and magnetization predictions.

For specific heat and magnetic susceptibility, all pipeline approaches correctly predict discontinuities near the critical temperature, capturing the characteristic shape of these observables across the temperature range. However, as shown in Figure 4, specific heat shows systematic overestimation across temperatures. This behavior reflects the model's tendency to generate configurations with slightly higher variability than the training dataset, which translates to enhanced thermal fluctuations in the predicted observables.

Notably, the model preserves the essential thermodynamic structure by correctly predicting both the functional form of the observables and the critical behavior, demonstrating that our approach successfully captures the underlying physical dynamics despite minor quantitative deviations in second-order moment.

## 6 CONCLUSION

In this paper we address the challenge of interpreting diffusion steps in flow-matching models by constraining trajectories to physically meaningful transitions. Our approach maps each vector field update to the equilibrium state of a physical process, transforming abstract flow evolution into concrete, physically-grounded dynamics.

We validate this framework through a 2D Ising model implementation with temperature-driven diffusion in latent space. The encoder-decoder architecture with differentiable projector preserves physical constraints while enabling continuous flow dynamics, where each update corresponds to a thermal equilibrium transition along the cooling schedule.

Experimental results across different lattice sizes demonstrate preserved physical fidelity with computational speedups over Monte Carlo methods, correctly capturing phase transitions and critical phenomena while achieving superior scaling for larger systems.

We hope that, by studying and applying this framework to different diffusion models, future work will guide us *step by step* toward unraveling the underlying mechanisms of how these models learn to diffuse, with implications for both model explainability and efficient architectural design.

## REPRODUCIBILITY STATEMENT

To ensure full reproducibility, we provide all source code, datasets, and experimental configurations in a public repository (Anonymous (2024)). Dataset generation employs Monte Carlo methods (Metropolis and Wolff algorithms) with documented convergence criteria and equilibration procedures, including comprehensive distribution analysis and statistical validation across all experimental conditions. All hyperparameters were empirically determined through iterative experimentation, with detailed parameter choices documented in the associated repository. Experiments were implemented in pure PyTorch without external deep learning frameworks and executed on a MacBook with Apple M3 chip utilizing integrated GPU and CPU resources.

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

## A  APPENDIX

### A.1  RESULTS ON DIFFERENT GRID SIZES

This section complements the main analysis by evaluating our model on $32 \times 32$ and $64 \times 64$ lattices, presenting qualitative spin configurations across temperatures and quantitative comparisons of thermodynamic observables with Monte Carlo results.

### A.1.1 DATASETS FOR 32 AND 64 GIRDS

Figures 5 and 6 illustrate that the datasets generated for grid sizes $N = 32$ and $N = 64$ are statistically equivalent to those produced with a $48 \times 48$ grid.

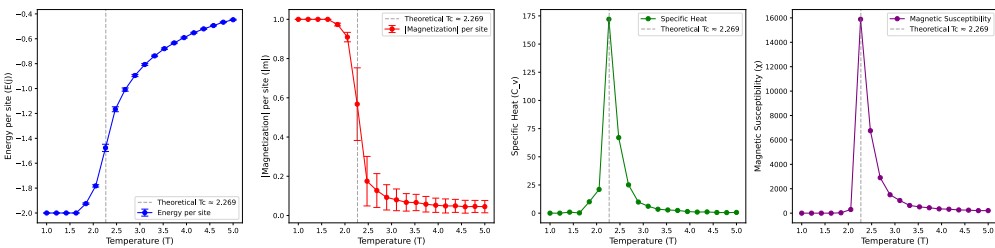

Figure 5: Observables ground truth (grid size $N = 32$)

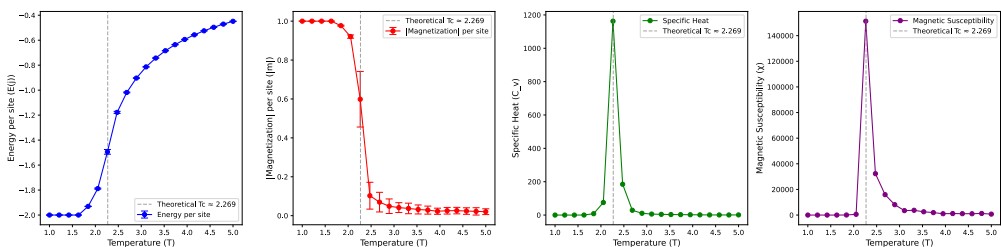

Figure 6: Observables ground truth (grid size $N = 64$)

### A.1.2 QUALITATIVE RESULTS

The qualitative plots allow visual inspection of the structural coherence of generated configurations, particularly near the critical temperature, where spatially organized spin clusters emerge. These samples, shown in Fig. 7 and Fig. 8, illustrate the model's ability to reproduce key structural features associated with phase transitions.

For the $32 \times 32$ (Fig. 7) and $64 \times 64$ (Fig. 8) grids, the same considerations discussed in the previous section apply.

### A.1.3 QUANTITATIVE RESULTS

The quantitative results, shown in Fig. 9 and Fig. 10, report for each temperature the values predicted by the model for four fundamental thermodynamic observables (energy, magnetization, specific heat, and magnetic susceptibility) and directly compare them to reference values from Monte Carlo simulations.

Analyzing the observable plots, we understand that the simple Monte Carlo method (MC−15) completely fails to capture the correct physical behavior, especially as the lattice size increases. The MH−15 method, while showing an energy trend roughly consistent with theoretical expectations at $32 \times 32$ (see Fig. 9), still produces inaccurate results for other observables. These discrepancies become more pronounced with fewer Monte Carlo steps and for larger grids, as show in Fig. 10.

In contrast, the decoder consistently learns the correct theoretical trends across all observables in both Fig. 9 and Fig. 10. Although the generated samples exhibit higher variance in energy and magnetization, leading to slightly overestimated magnetization and magnetic susceptibility, the predictions still align with the expected positions of the discontinuities.

### A.2 ONSAGER EXACT SOLUTION

For the curious reader, since our analysis relies heavily on the Ising model's predictable and closed-form expressions for observables, we include Onsager's free energy derivation as foundational

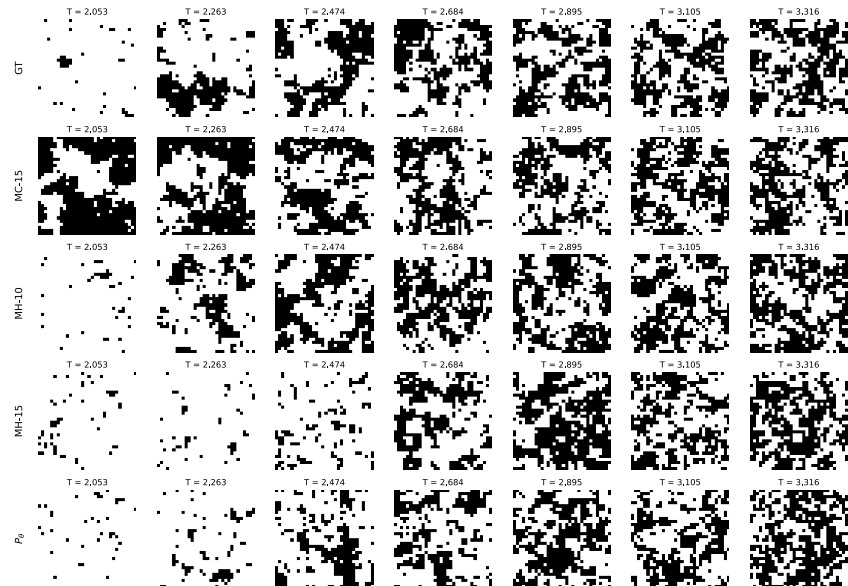

Figure 7: Comparison of decoding methods in the encoder-vector field pipeline across increasing temperature values (left to right) for grid size $N = 32$.

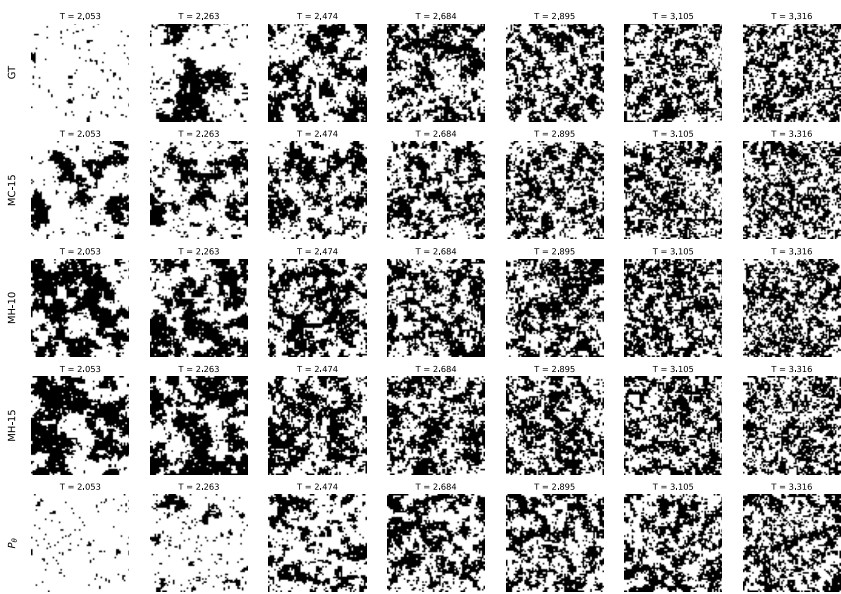

Figure 8: Comparison of decoding methods in the encoder-vector field pipeline across increasing temperature values (left to right) for grid size $N = 64$.

reference for our computational framework. The following section heavily relies on Ridderstolpe (2017), Onsager (1944) studies.

The square lattice model is examined through its partition function, expressed via the transfer matrices $V$ and $W$.

The star-triangle relation establishes commutation properties among transfer matrices. Combined with operators $R$, $C$, and Kramers-Wannier duality, this yields eigenvalue relations parametrized by elliptic functions for free energy computation.

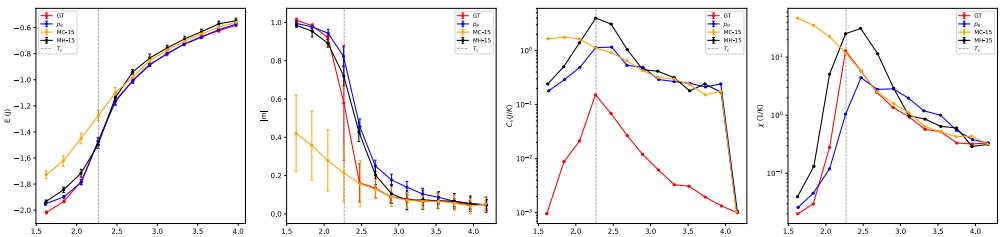

Figure 9: Comparison of physical observables defined over the cooling schedule (grid size $N = 32$)

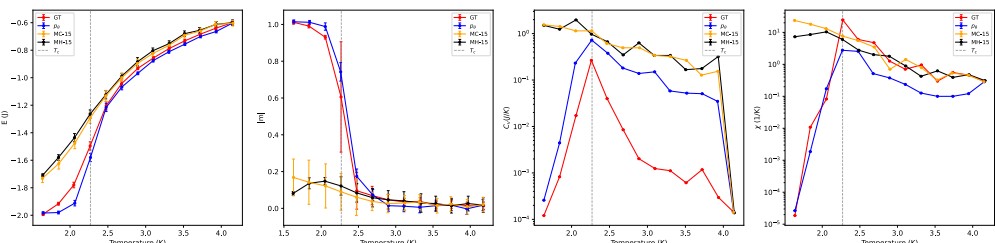

Figure 10: Comparison of physical observables defined over the cooling schedule (grid size $N = 64$)

Critical phenomena and phase transitions are identified through Kramers-Wannier duality analysis and free energy singularities.

### A.2.1 PARTITION FUNCTION DERIVATION

We analyze an $N$-site square lattice system without external fields, implementing periodic boundary conditions that create a toroidal topology (Figure 11).

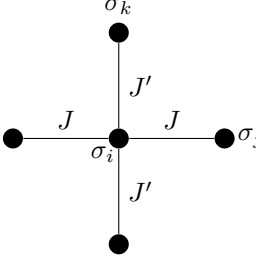

Figure 11: Description of Nearest-neighbour interaction in one site. Image taken from Ridderstolpe (2017).

The model incorporates nearest-neighbor interactions with anisotropic coupling constants: horizontal interactions characterized by strength $J$ and vertical interactions by $J'$.

The Hamiltonian is constructed from pairwise interaction terms:

$$H(\sigma) = -J \sum_{\langle i,j \rangle_h} \sigma_i \sigma_j - J' \sum_{\langle i,k \rangle_v} \sigma_i \sigma_k \tag{7}$$

where the summations extend over horizontal and vertical nearest-neighbor pairs respectively.

The corresponding partition function takes the form:

$$Z = \sum_{\{\sigma\}} \exp \left[ \beta J \sum_{\langle i,j \rangle_h} \sigma_i \sigma_j + \beta J' \sum_{\langle i,k \rangle_v} \sigma_i \sigma_k \right] \tag{8}$$

with dimensionless parameters $K = \beta J$ and $L = \beta J'$.

### A.2.2 TRANSFER MATRIX FORMALISM

The two-dimensional lattice can be represented through a matrix formulation by organizing the system into rows and establishing transfer relationships between adjacent layers. The lattice structure is reoriented to facilitate row-based analysis (Figure 12).

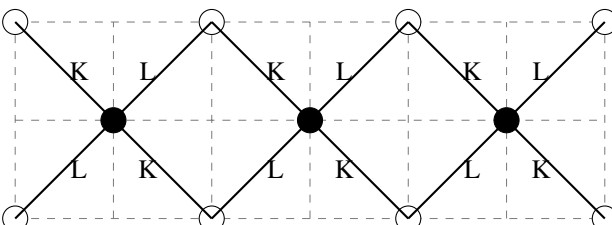

Figure 12: Square lattice configuration with rotated orientation. Image taken from Ridderstolpe (2017).

Let $m$ denote the total number of rows, with each row containing $n$ spin sites. The spin configuration of row $r$ is defined as $\phi_r$, where $1 \leq r \leq m$ and $\phi_r = \{\sigma_1, \sigma_2, \ldots, \sigma_n\}$.

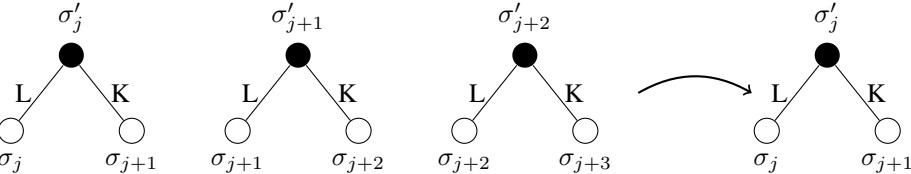

Figure 13: The interaction between two rows. Image taken from Ridderstolpe (2017).

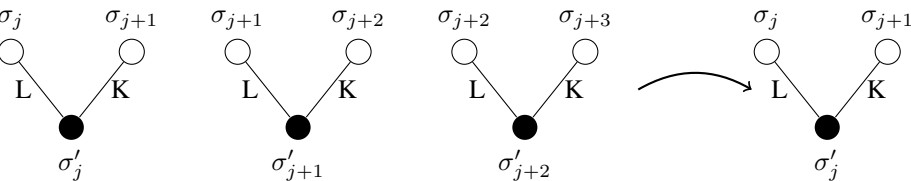

Figure 14: The interaction between two rows is described as a repetition of the right side interactions, we note the interaction share the middle row Figure 13. Image taken from Ridderstolpe (2017).

Inter-row interactions are examined through nearest-neighbor bonds between adjacent rows. For two consecutive rows with lower configuration $\phi$ and upper configuration $\phi'$, each site $\sigma_j$ in the lower row couples with specific sites in the upper row according to the lattice geometry.

The coupling between rows is characterized by Boltzmann weights. For the first type (Figure 13) of row-to-row interaction:

$$V(\phi, \phi') = \exp\left[\sum_{j=1}^{n}(K\sigma_{j+1}\sigma'_j + L\sigma_j\sigma'_j)\right]$$

For the alternate row (Figure 14) coupling pattern:

$$W(\phi, \phi') = \exp\left[\sum_{j=1}^{n}(K\sigma_j\sigma'_j + L\sigma_j\sigma'_{j+1})\right]$$

The total partition function becomes a product over these row-coupling weights:

$$Z_N = \sum_{\phi_1}\sum_{\phi_2}\cdots\sum_{\phi_m} V(\phi_1, \phi_2)W(\phi_2, \phi_3)\cdots V(\phi_{m-1}, \phi_m)W(\phi_m, \phi_1)$$

where periodic boundary conditions are applied to the final weight term.

Matrices $V$ and $W$ are constructed with elements $V_{\phi,\phi'}$ and $W_{\phi,\phi'}$ corresponding to each possible row configuration pair. Since each row admits $2^n$ possible spin configurations, both matrices have dimensions $2^n \times 2^n$.

The partition function can be rewritten using matrix notation:

$$Z_N = \sum_{\phi_1} \sum_{\phi_2} \cdots \sum_{\phi_m} V_{\phi_1,\phi_2} W_{\phi_2,\phi_3} \cdots W_{\phi_m,\phi_1} = \sum_{\phi_1} (VWV \cdots W)_{\phi_1,\phi_1}$$

$$= \sum_{\phi_1} ((VW)^{m/2})_{\phi_1,\phi_1} = \mathrm{Tr}\left[ (VW)^{m/2} \right]$$

Since the trace of any matrix equals the sum of its eigenvalues:

$$Z_N = \Lambda_1^m + \Lambda_2^m + \cdots + \Lambda_{2^n}^m$$

where $\Lambda_1, \Lambda_2, \ldots, \Lambda_{2^n}$ represent the eigenvalues of the product $VW$.

### A.2.3    STAR-TRIANGLE TRANSFORMATION AND COMMUTATION RELATIONS

The commutation properties of transfer matrices $V$ and $W$ are established by examining their functional dependence on coupling parameters. Let $V(K, L)$ and $W(K', L')$ denote transfer matrices with arbitrary complex coupling constants. The investigation focuses on whether these matrices satisfy the commutation relation:

$$V(K,L)W(K',L') = V(K',L')W(K,L) \tag{9}$$

The elements of the product $V(K, L)W(K', L')$ are given by:

$$(VW)_{\phi,\phi'} = \sum_{\phi''} V_{\phi,\phi''} W_{\phi'',\phi'}$$

Substituting the explicit forms:

$$(VW)_{\phi,\phi'} = \sum_{\phi''} \exp\left[ \sum_{j=1}^{n} (K\sigma_{j+1}\sigma_j'' + L\sigma_j\sigma_j'') \right] \times \exp\left[ \sum_{j=1}^{n} (K'\sigma_j''\sigma_j' + L'\sigma_j''\sigma_{j+1}') \right]$$

Combining the exponentials:

$$(VW)_{\phi,\phi'} = \sum_{\phi''} \prod_{j=1}^{n} \exp\left[ \sigma_j''(K\sigma_{j+1} + L\sigma_j + K'\sigma_j' + L'\sigma_{j+1}') \right] \tag{10}$$

Since $\sigma_j'' = \pm 1$, each factor in the product becomes:

$$\sum_{\sigma_j''=\pm 1} \exp\left[ \sigma_j''(K\sigma_{j+1} + L\sigma_j + K'\sigma_j' + L'\sigma_{j+1}') \right]$$

$$= 2\cosh(K\sigma_{j+1} + L\sigma_j + K'\sigma_j' + L'\sigma_{j+1}')$$

Therefore:

$$(VW)_{\phi,\phi'} = \prod_{j=1}^{n} X(\sigma_j, \sigma_{j+1}; \sigma_j', \sigma_{j+1}')$$

where:

$$X(\sigma_j, \sigma_{j+1}; \sigma_j', \sigma_{j+1}') = 2\cosh(K\sigma_{j+1} + L\sigma_j + K'\sigma_j' + L'\sigma_{j+1}')$$

The function $X$ describes a four-vertex star interaction. To establish commutation, the star-triangle transformation is employed. Consider a three-vertex system with interaction:

$$w(\sigma_i, \sigma_j, \sigma_k) = \sum_{\sigma_l} \exp[\sigma_l(B_1\sigma_i + B_2\sigma_j + B_3\sigma_k)] = 2\cosh(B_1\sigma_i + B_2\sigma_j + B_3\sigma_k)$$

This star interaction can be transformed to a triangle interaction:

$$w(\sigma_i, \sigma_j, \sigma_k) = R \exp[C_1 \sigma_i \sigma_j + C_2 \sigma_j \sigma_k + C_3 \sigma_k \sigma_i]$$

The star-triangle relations connect these representations by evaluating for all spin configurations:

$$w(+,+,+) = 2\cosh(B_1 + B_2 + B_3) = R\exp[C_1 + C_2 + C_3]$$
$$w(+,+,-) = 2\cosh(B_1 + B_2 - B_3) = R\exp[C_1 - C_2 - C_3]$$
$$w(-,+,+) = 2\cosh(-B_1 + B_2 + B_3) = R\exp[-C_1 - C_2 + C_3]$$
$$w(+,-,+) = 2\cosh(B_1 - B_2 + B_3) = R\exp[-C_1 + C_2 - C_3]$$

To determine commutation conditions, the four-vertex interaction is augmented with an additional coupling $M\sigma_j \sigma_j'$:

$$w_1(\sigma_j, \sigma_{j+1}; \sigma_j', \sigma_{j+1}') = \sum_{\sigma_j''} \exp[M\sigma_j \sigma_j' + \sigma_j''(L\sigma_j + K\sigma_{j+1} + L'\sigma_{j+1}' + K'\sigma_j')]$$

$$= e^{M\sigma_j \sigma_j'} \sum_{\sigma_j''} \exp[\sigma_j''(L\sigma_j + K\sigma_{j+1} + L'\sigma_{j+1}' + K'\sigma_j')]$$

For the interchanged case:

$$w_2(\sigma_j, \sigma_{j+1}; \sigma_j', \sigma_{j+1}') = e^{M\sigma_{j+1}\sigma_{j+1}'} \sum_{\sigma_j''} \exp[\sigma_j''(L'\sigma_{j+1}' + K'\sigma_j' + L\sigma_j + K\sigma_{j+1})]$$

Commutation requires $w_1 = w_2$. The triangle interaction constants are identified as:

$$C_1 = L, \quad C_2 = K', \quad C_3 = M$$

Applying the star-triangle transformation yields the constraint:

$$B_1 = L', \quad B_3 = K$$

Inserting into the star-triangle relations and solving:

$$2\cosh(L' + B_2 + K) = R\exp[L + K' + M]$$
$$2\cosh(L' + B_2 - K) = R\exp[L - K' - M]$$
$$2\cosh(-L' + B_2 + K) = R\exp[-L + K' - M]$$
$$2\cosh(L' - B_2 + K) = R\exp[-L - K' + M]$$

Adding and subtracting these equations systematically:

$$4\cosh(B_2)\cosh(K + L') = 2Re^M \cosh(L + K')$$
$$4\sinh(B_2)\sinh(K + L') = 2Re^M \sinh(L + K')$$
$$4\cosh(B_2)\cosh(K - L') = 2Re^{-M} \cosh(L - K')$$
$$4\sinh(B_2)\sinh(K - L') = 2Re^{-M} \sinh(L - K')$$

Taking ratios and multiplying:

$$\frac{\cosh(K + L')\sinh(K - L')}{\cosh(K - L')\sinh(K + L')} = \frac{\cosh(L + K')\sinh(L - K')}{\sinh(L + K')\cosh(L - K')}$$

This simplifies to the fundamental commutation condition:

$$\sinh(2K)\sinh(2L) = \sinh(2K')\sinh(2L')$$

When coupling constants satisfy the constraint equation, the transfer matrices commute:

$$V(K, L)W(K', L') = W(K', L')V(K, L)$$

This commutation property is independent of the auxiliary parameter $M$, which can be set to zero for convenience. The constraint $\sinh(2K)\sinh(2L) = \sinh(2K')\sinh(2L')$ defines the manifold in parameter space where exact solutions can be constructed.

### A.2.4 FUNCTIONAL RELATION OF THE EIGENVALUES

A functional equation governing the eigenvalues of the transfer matrix product is established by exploiting the commutation properties and operator relations developed in previous sections. Since the transfer matrices $V(K, L)$ and $W(K, L)$ commute with themselves and with operators $R$ and $C$ when the coupling constants satisfy $\sinh(2K)\sinh(2L) = \text{constant}$, all these matrices share common eigenvectors. Let $x(k)$ denote a common eigenvector with corresponding eigenvalues $v(K, L)$, $c$, and $r$ for matrices $V(K, L)$, $C$, and $R$ respectively:

$$V(K, L)x(k) = v(K, L)x(k)$$
$$Cx(k) = cx(k)$$
$$Rx(k) = rx(k)$$

The periodic boundary conditions impose constraints on the eigenvalues. Since $C^n = I$ and $R^2 = I$, the relations $c^n = r^2 = 1$ hold. From the matrix inversion relation:

$$V(K, L)V\left(L + \frac{i\pi}{2}, -K\right)C = (2i\sinh 2L)^n I + (-2i\sinh 2K)^n R \tag{11}$$

Application to the common eigenvector $x(k)$ yields:

$$v(K, L)v\left(L + \frac{i\pi}{2}, -K\right)c = (2i\sinh 2L)^n + (-2i\sinh 2K)^n r \tag{12}$$

The eigenvalues of the transfer matrix product $VW$ are related through $V(K, L)W(K, L) = V^2(K, L)C$, giving $\Lambda^2(K, L) = v^2(K, L)c$ and thus $\Lambda(K, L) = v(K, L)c^{1/2}$. Substitution into the functional equation gives:

$$\Lambda(K, L)\Lambda\left(L + \frac{i\pi}{2}, -K\right) = (2i\sinh 2L)^n + (-2i\sinh 2K)^n r$$

Using the constraint $k = (\sinh 2K \sinh 2L)^{-1}$ and parametrization $\sinh 2K = x$ and $\sinh 2L = (kx)^{-1}$, this becomes:

$$\Lambda(u)\Lambda(u + I') = \left(-\frac{2}{k\text{sn}(iu)}\right)^{2p} + (-2\text{sn}(iu))^{2p}r$$

where the elliptic parameter $u$, half-period $I'$, $x = -i\text{sn}(iu)$, and $n = 2p$ have been introduced. This functional relation completely determines the eigenvalue spectrum and forms the foundation for calculating the free energy.

### A.2.5 GENERAL EXPRESSION FOR THE EIGENVALUES

A complete analytical form for the eigenvalues is derived by developing a systematic approach that connects the transfer matrix structure to elementary functions. Starting from the functional relation derived earlier, eigenvalue expressions are sought that can be evaluated without explicit recourse to elliptic function computations.

The key insight involves parametrizing the eigenvalue problem using angular variables that naturally arise from the periodic structure of the lattice. A set of characteristic angles is defined as:

$$\theta_j = \begin{cases} \frac{\pi}{2p}\left(j - \frac{1}{2}\right) & \text{for } r = 1 \\ \frac{\pi j}{2p} & \text{for } r = -1 \end{cases}$$

where $j = 1, 2, \ldots, 2p$ and $r = \pm 1$ distinguishes between the two eigenvalue branches.

For each angle $\theta_j$, auxiliary coefficients are constructed:

$$c_j = k^{-1}(1 + k^2 - 2k\cos(2\theta_j))^{1/2}$$

These coefficients encode the geometric constraints imposed by the lattice topology and the commutation relations between transfer matrices.

The eigenvalue components are expressed through rational functions:

$$u_j = \frac{\cosh 2K \cosh 2L + c_j}{\exp(i\theta_j)\sinh 2K + \exp(-i\theta_j)\sinh 2L}$$

where the denominator captures the interference between horizontal and vertical coupling strengths.

The complete eigenvalue spectrum takes the factorized form:

$$\Lambda^2 = \tau(-4)^p[(\sinh 2L)^{2p} + r(\sinh 2K)^{2p}] \prod_{j=1}^{2p}(u_j)^{\gamma_j} \tag{13}$$

where the sign factor is:

$$\tau = \begin{cases} +1 & \text{for } r = 1 \\ -i & \text{for } r = -1 \end{cases}$$

and $\gamma_j = \pm 1$ are determined by maximization conditions.

For physical systems with real positive coupling constants, the constraint $|u_j| \geq 1$ ensures that the dominant eigenvalue corresponds to $\gamma_j = +1$ for all $j$. This leads to the simplified maximum eigenvalue:

$$\Lambda_{\max}^2 = \prod_{j=1}^{2p} 2(\cosh 2K \cosh 2L + c_j)$$

This expression provides a direct computational pathway for evaluating the partition function and subsequently the free energy, circumventing the need for specialized function evaluations while maintaining complete analytical precision.

### A.2.6 Maximum Eigenvalue and the Free Energy

The free energy is determined by identifying the dominant eigenvalue in the thermodynamic limit and establishing its connection to the partition function. In systems with large numbers of degrees of freedom, the largest eigenvalue overwhelmingly dominates the trace, allowing extraction of the thermodynamic properties.

The partition function for the square lattice system takes the form:

$$Z_N = \Lambda_1^m + \Lambda_2^m + \cdots + \Lambda_{2^n}^m$$

where $m$ represents the number of rows and $\Lambda_i$ are the eigenvalues of the transfer matrix product $VW$. In the thermodynamic limit where $m \to \infty$, the term with the largest absolute eigenvalue dominates:

$$Z_N \approx (\Lambda_{\max})^m$$

To identify the maximum eigenvalue, the general eigenvalue expression is examined. The eigenvalues depend on the choice of parameters $r = \pm 1$ and signs $\gamma_j = \pm 1$. For real positive coupling constants $K$ and $L$, the constraint analysis shows that $|u_j| \geq 1$ for all $j$, where:

$$u_j = \frac{\cosh 2K \cosh 2L + c_j}{\exp(i\theta_j)\sinh 2K + \exp(-i\theta_j)\sinh 2L}$$

The maximum eigenvalue is achieved when all $\gamma_j = +1$ and corresponds to the choice $r = 1$. The Perron-Frobenius theorem guarantees that for matrices with positive entries, the largest eigenvalue is real and positive with a corresponding eigenvector having strictly positive components. This condition is satisfied only when $r = 1$.

With $r = 1$, the angles become $\theta_j = \frac{\pi}{2p}(j - \frac{1}{2})$ for $j = 1, 2, \ldots, 2p$. The maximum eigenvalue takes the form:

$$\Lambda_{\max}^2 = \prod_{j=1}^{2p} 2(\cosh 2K \cosh 2L + c_j)$$

Taking the logarithm yields:

$$\ln \Lambda_{\max} = \frac{1}{2} \sum_{j=1}^{2p} \ln[2(\cosh 2K \cosh 2L + c_j)]$$

The free energy per lattice site is obtained from the relation $F = -k_B T \ln Z_N$. With the total number of sites $N = m \times 2p$ and using $Z_N \approx (\Lambda_{\max})^m$:

$$f = -\frac{k_B T}{2p} \ln \Lambda_{\max}$$

Substituting the eigenvalue expression:

$$f = -\frac{k_B T}{4p} \sum_{j=1}^{2p} \ln[2(\cosh 2K \cosh 2L + c_j)]$$

The function is defined as:

$$F(\theta) = \ln[2(\cosh 2K \cosh 2L + k^{-1}(1 + k^2 - 2k \cos 2\theta)^{1/2})]$$

For large $p$, the discrete sum can be approximated by an integral with interval length $\frac{\pi}{2p}$:

$$f = -\frac{k_B T}{2\pi} \int_0^\pi F(\theta) d\theta \tag{14}$$

This integral representation provides the exact free energy per lattice site for the square lattice Ising model. The integrand $F(\theta)$ encodes all the essential physics of the system, including the anisotropy effects and the approach to criticality. The free energy exhibits the expected analytical properties and reduces to known limits in appropriate parameter regimes.

### A.2.7 CRITICAL TEMPERATURE AND PHASE TRANSITION

The critical temperature is identified through analysis of free energy singularities and application of the Kramers-Wannier duality principle. Phase transitions manifest as non-analytical behavior in thermodynamic quantities at specific parameter values.

The free energy expression:

$$f = -\frac{k_B T}{2\pi} \int_0^\pi F(\theta) d\theta$$

where:

$$F(\theta) = \ln[2(\cosh 2K \cosh 2L + k^{-1}(1 + k^2 - 2k \cos 2\theta)^{1/2})]$$

Singularities are located by examining the integrand behavior as functions of parameter $k = (\sinh 2K \sinh 2L)^{-1}$. Using relations $K = J/(k_B T)$ and $L = J'/(k_B T)$:

$$k = (\sinh(2J/k_B T) \sinh(2J'/k_B T))^{-1}$$

The integrand decomposes as:

$$F(\theta) = \ln[2 \cosh 2K \cosh 2L] + \ln\left[1 + \frac{k^{-1}(1 + k^2 - 2k \cos 2\theta)^{1/2}}{\cosh 2K \cosh 2L}\right]$$

The first term is analytical everywhere; singularities arise from the second term. For logarithmic expansion $\ln(1 + x) = x - \frac{x^2}{2} + \frac{x^3}{3} + \cdots$, the critical term is:

$$\frac{k^{-1}(1 + k^2 - 2k \cos 2\theta)^{1/2}}{\cosh 2K \cosh 2L}$$

Singularities occur when this expression becomes non-analytical. The square root factor $(1 + k^2 - 2k\cos 2\theta)^{1/2}$ develops branch point behavior at $k = 1$ when $\cos 2\theta = 1$.

The singular contribution to free energy:

$$f_s = -\frac{k_B T}{2\pi} \int_0^\pi \frac{k^{-1}(1 + k^2 - 2k\cos 2\theta)^{1/2}}{\cosh 2K \cosh 2L} d\theta$$

Through integral analysis and expansion around $k = 1$:

$$f_s = -\frac{k_B T(1 + k)(1 - k)}{4\pi k \cosh 2K \cosh 2L} \ln\left(\frac{1 + k}{1 - k}\right) \tag{15}$$

The singularity manifests as $k \to 1$, where the logarithmic term diverges. This occurs when:

$$\sinh(2J/k_B T_c)\sinh(2J'/k_B T_c) = 1$$

For the isotropic case $J = J'$:

$$\sinh^2(2J/k_B T_c) = 1 \tag{16}$$

yielding the critical temperature:

$$k_B T_c = \frac{2J}{\ln(1 + \sqrt{2})} \approx 2.269J \tag{17}$$

### A.2.8 INTERNAL ENERGY

The internal energy per site is obtained by differentiating the free energy with respect to temperature. The thermodynamic relation:

$$u = -\frac{\partial}{\partial \beta}(\beta f)$$

where $\beta = (k_B T)^{-1}$. From the free energy expression and noting that $\frac{\partial K}{\partial \beta} = J$ and $\frac{\partial L}{\partial \beta} = J'$:

$$u = -J\frac{\partial f}{\partial K} - J'\frac{\partial f}{\partial L}$$

For the isotropic case $J = J'$, the internal energy becomes:

$$u = -2J \coth(2K)\left[1 + \frac{2}{\pi}(2\tanh^2(2K) - 1)\mathcal{K}(k^*)\right] \tag{18}$$

where $\mathcal{K}(k^*)$ is the complete elliptic integral of the first kind with modulus $k^* = 2\sinh(2K)/\cosh^2(2K)$. At the critical temperature, the internal energy exhibits a logarithmic singularity characteristic of the two-dimensional Ising model.

