# OpenReview forum: "On the Flow Matching Interpretability"
_ICLR.cc/2026/Conference — ICLR 2026 Conference Desk Rejected Submission_

### Official Review · Reviewer_eFi1 · 2025-10-20

**Soundness:** 2
**Presentation:** 3
**Contribution:** 1
**Rating:** 2
**Confidence:** 3

**Summary:**

## Summary

This paper proposes a framework for improving interpretability in flow-matching
generative models by constraining trajectories to physically meaningful equilibrium
states. The authors show that macroscopic observables (energy, magnetization, specific
heat, susceptibility) reproduce expected phase-transition behavior and report speedups
over Monte Carlo at larger lattice sizes. The core idea is interesting and the
execution within the Ising domain is reasonably thorough. However, the scope is too narrow, the interpretability claim is not
rigorously validated, and critical theoretical questions about probability flow remain
unanswered. The paper would be significantly strengthened by: (1) rigorous
interpretability metrics, (2) analysis of probability conservation, (3) expanded
evaluation and ablations, (4) stronger baselines, and (5) demonstration on additional
systems. In its current form, the contribution is insufficient for a top-tier venue.

**Strengths:**

1. **Novel perspective on interpretability**: The paper addresses a genuine gap in
   generative modeling—intermediate steps in diffusion/flow models lack clear semantic
   meaning. Constraining trajectories to physical equilibrium states is a creative
   solution.

2. **Clear presentation**: The encoder-latent field-projector pipeline is well-described
   and easy to follow. The piecewise linear interpolation strategy is sensible.

**Weaknesses:**

### Major Issues

1. **Extremely limited scope**: Evidence is restricted to a single, well-studied toy
   system (2D Ising with macroscopic metrics). No other physical systems, no real data,
   no standard ML benchmarks. This severely limits the impacts of the results, the
   paper claims a "general framework" but demonstrates only a single instantiation.

2. **Interpretability is not rigorously measured**: The paper defines interpretability by
   construction (alignment with β) but provides no quantitative metrics. What makes these
   trajectories actually interpretable? Proposed alternatives:
   - Path faithfulness: Do small latent perturbations produce predictable physical changes?
   - Counterfactual consistency: Can interventions in latent space be validated against physics?

   Without such metrics, "interpretability" remains a claim rather than a validated property.

3. **Probability flow and marginal correctness not addressed**: The paper composes a
   continuous latent ODE with a non-differentiable projector/MC refinement but does not
   analyze:
   - Does ∇·v_θ satisfy expected divergence properties? (I.e, continuity equation)
   -  Do induced marginals at intermediate β match target
     distributions (both in latent space and after projection)?
   -  Can the authors justify that the projector preserves probability
     mass and maintains the flow-matching guarantees?

   As written, the projector can remap probability arbitrarily, potentially breaking the
   theoretical foundations of the approach. No diagnostics or proofs are provided.

4. The authors themselves acknowledge systematic overestimation of C_v but do not investigate
   whether this reflects issues in the vector field or decoder.

5. **No comparison to relevant baselines**: Comparisons are primarily against MC variants.
   Missing: head-to-head evaluation against β-conditioned diffusion/FM baselines, or
   physics-aware loss functions. Without such comparisons, it's unclear whether the
   constraint actually improves quality/semantics over standard FM.

### Minor Issues

6. **Incomplete ablations**: The method depends on MC-generated schedules, number of
   transitions K, KDE bandwidths, and projector architecture. No ablation studies on
   these design choices—unclear which components are critical and how sensitive the
   method is to hyperparameters.

7. **Systematic Cv overestimation unexplained**: The paper notes this but does not
   identify the source (vector field vs. decoder) or investigate mitigation strategies.

8. **No end-to-end cost accounting**: Total cost includes MC data generation, training,
   and inference. Speedup claims focus only on inference time; full pipeline cost vs.
   pure MC sampling alone is not provided.

## Suggestions for Improvement

- **Add rigorous interpretability metrics**: Measure path faithfulness, counterfactual
  sensitivity, or semantic consistency to validate the interpretability claim.

- **Analyze probability flow**: Provide diagnostics for continuity-equation compliance
  and marginal correctness, or provide theoretical justification for why violations are
  tolerable.

- **Expand evaluation**: Include two-point correlations, structure factors, and
  correlation length. Perform ablation studies on β-grid density, K, KDE bandwidth,
  and projector capacity.

- **Strengthen baselines**: Compare against β-conditioned diffusion/FM baselines and
  report full end-to-end costs.

- **Broaden scope**: Demonstrate the framework on at least one additional physical
  system or real measurement data. Temper "generality" claims to match current scope.

## Minor Comments

- Provide exact hyperparameters (optimizer, learning rate, model sizes, KDE bandwidths,
  seeds) for full reproducibility.

- Include spatial visualizations near T_c with correlation overlays to qualitatively
  assess structural fidelity.

- Clarify whether the C_v overestimation stems from the vector field or projector via
  ablation with a frozen projector.

**Questions:**

1. **Probability preservation**: Can you provide diagnostics quantifying continuity-equation
   compliance along the latent trajectory (e.g., mean divergence of v_θ, score proxy
   residuals)? Do marginals at intermediate β match the target Gibbs distributions?

2. **Microstate-level fidelity**: Beyond macroscopic observables, can you measure
   two-point correlations, correlation length, and cluster-size distributions near T_c
   to validate that the learned trajectory preserves fine-grained structure?

3. **Projector justification**: How do you ensure the projector does not disrupt mass
   conservation? What is its role—is it primarily error correction or does it serve a
   fundamental purpose in the pipeline?

4. **Baseline comparisons**: How does performance compare to β-conditioned diffusion
   baselines or FM models with physics-aware losses? What is the actual speedup relative
   to total pipeline cost?

5. **Sensitivity analysis**: How do results vary with β-grid resolution, number of stored
   transitions K, KDE bandwidth, and projector capacity? Which design choices are most
   critical?

6. **Generalizability**: Can you demonstrate results on at least one additional system
   (e.g., Potts model, XY model, or simple PDE data)? This would strengthen claims about
   the framework's generality.

---

### Official Review · Reviewer_VwBZ · 2025-10-26

**Soundness:** 1
**Presentation:** 2
**Contribution:** 1
**Rating:** 2
**Confidence:** 4

**Summary:**

The paper aims to address the gap in interpretability of intermediate steps in the flow-based generation process. The proposed approach compares the flow parameter $t$ with the temperature of a 2D Ising model. The distribution $p_t$ at certain discrete steps in $t$ is constrained to follow the Ising model at the corresponding temperature. Hence, the flow becomes a piecewise linear trajectory.

The proposed method is compared with the Metropolis-Hastings (MH) algorithm for different lattice sizes. The results show faster sampling than MH and a better match of statistics (expected values) to the ground truth.

**Strengths:**

The paper is well written. The preliminaries are explained very well, and the write-up on the Ising model is very good.

**Weaknesses:**

I believe that the approach is not generalisable to target distributions without a scalar parameter; e.g., to the standard problem of image generation, or to other lattice systems that have more than one scalar parameter.

The paper begins with the aim of making flow matching interpretable; however, the results do not discuss interpretability. They discuss sampling speed and sample statistics. I am unsure whether the proposed method contributes to interpretability itself.

In the experiments, the baselines are weak. There are many generative models (GANs, VAEs, NFs, and even flow matching models) existing in the literature, even for lattice systems (as already cited by the authors). Even if the sampling speed and sample statistics are to be assessed, why not compare with those methods?

**Questions:**

- Please answer my concerns mentioned in the weaknesses section, if you disagree with them.
- Flow matching aims at creating trajectories that are one-to-one or bijective maps from $x_0$ to $x_1$, both of which are in the continuous space. But in the case of the Ising model, x lies in a discrete space. How does the proposed method ensure the bijectivity of trajectories?
- Moreover, the number of probable discrete states changes with temperature, as the entropy changes. The extreme case is zero temperature, where only 2 states are probable. How is bijective mapping done then?
- What is the starting distribution $p_0$? Does it correspond to the Ising model at some temperature? By definition, $p_0$ should be easy to sample.
- The training costs are not discussed. The common flow matching training averages over random pairs $(x_0,x_1)$. Does the proposed method average over random sequences $(x_0, ..., x_\beta, ..., x_1)$? I fear that the computational costs may increase significantly.
- How does the general flow matching, where only $x_0$ and $x_1$ are used to construct flow vector fields, compare with the proposed "piecewise" linear trajectories? An ablation study could help.

I have some questions about writing. Answering them will help to better understand.
- In the experiments, does the ground truth refer to the Wolff algorithm?
- What are the values of $t$ used, and what are the corresponding values of $beta$?

---

### Official Review · Reviewer_WDHT · 2025-10-27

**Soundness:** 2
**Presentation:** 3
**Contribution:** 1
**Rating:** 2
**Confidence:** 5

**Summary:**

A flow matching model is trained on continuous latent representations of the states of a 2D Ising model, with the time variable mapped to the inverse temperature $\beta$ of the system. The goal is to make a flow where the intermediate steps are interpretable. Experiments at various lattice sizes are shown.

**Strengths:**

1. The paper is generally well written and easy to understand.
2. The experiments demonstrate that the flow approach amortizes the cost of drawing equilibrium samples, since expensive sampling is performed only once during training.

**Weaknesses:**

Early in the paper, the authors motivate their work by pointing out that the intermediate steps of a flow matching model are not interpretable. It is also claimed around line 77 that 'This interpretability extends beyond the specific physical system, providing a general methodology for constraining generative flows to meaningful intermediate representations.' But it appears to me that the overall structure of your pipeline is similar to many examples of NN-amortized MCMC that are already present in the literature. See [1] and the references therein. The main novelty here, as I see it, is that you chose to model a _sequence_ of equilibrium states, each one at a different temperature/flow time. This is a restrictive class of problems, is it not? In other words, the lack of interpretability of intermediate steps is not a drawback of flow matching method itself, but a reflection of the problem to which it is applied.

Delving further into the details, the discussion till Sec. 4.2 is a recapitulation of well-known facts about flow-matching and the 2D Ising model. A sizable chunk of the appendix is also dedicated to the latter. While the experiments and the demonstrated inference speed-ups are appreciated, comparable amortized sampling techniques have been applied to a wide range of problems with similar benefits. Overall, the contributions appear incremental relative to existing work, and the paper does not present sufficiently novel advances to meet the typical standard of acceptance at ICLR.

**Questions:**

Not a question, but a small comment about formatting: the bibliography citations in the main text were not rendered as clickable links.

[1] Cabezas, A., Sharrock, L. & Nemeth, C. (2024). Markovian Flow Matching: Accelerating MCMC with Continuous Normalizing Flows. arXiv preprint arXiv:2405.14392.

---

### Official Review · Reviewer_giQj · 2025-11-01

**Soundness:** 2
**Presentation:** 3
**Contribution:** 1
**Rating:** 2
**Confidence:** 5

**Summary:**

The work proposes a flow-matching training procedure that learns transitions between Ising-model microstates generated at two nearby temperatures, $T_{n}$ and $T_{n+1}$, on a grid between $T_{\min}$ and $T_{\max}$. The overall transition from $T_{\min}$ to $T_{\max}$ is implemented as a piecewise linear flow matching over 20 preselected inverse-temperature points $\beta$. The timestep in the flow matching is then interpreted as temperature. The paper describes the necessary steps of embedding the discrete Ising grids into continuous latent representations and implements a decoding back into discrete space.

**Strengths:**

The paper is clearly written and the proposed approach is clearly presented.

The Ising setup features ground true solution and has many well-studied properties, as an analytical form of probability density distribution and the normalization constant.

Bridging generative modeling and statistical physics is an interesting research direction.

**Weaknesses:**

The motivation of the selected approach is not clear.

The described numerical experiments and conclusions are not applicable to other pretrained flow matching models.

**Questions:**

Why is the piecewise approach selected for distributions between $T_{\min}$ and $T_{\max}$? A more reasonable option is to train a standard, unconstrained flow-matching model directly between the $T_{\max}$ and $T_{\min}$ distributions and then compare the learned path with the ground-truth transition. This approach—by varying flow-matching parameters (schedule, interpolant, etc.)—would allow one to study the proximity between the learned trajectories and the physical ground-truth behavior near the $T_c$ point. One could, for example, compare the entropy of the learned intermediate distributions with the known entropy of the Ising model.

How many Metropolis–Hastings steps were performed during dataset generation for each lattice size? Why is this number of steps sufficient to reach thermodynamic equilibrium?

How is the quality of the learned encoder and decoder evaluated? In the MH decoder approach, a physical simulation (several steps of Metropolis–Hastings) is used, which injects explicit knowledge about the target distribution into the reverse sampling dynamics. The problem is that this MH simulation could produce physically meaningful states even if the learned vector field makes arbitrary predictions. In other words, the effects of the decoder and the learned vector field are not separated, so one cannot conclude that the proposed procedure actually learns to generate the correct samples.

---

### Author Response · Authors · 2025-11-22
**Responce to the reviews**

First of all, we would like to thank all the reviewers, as everything they wrote was valuable, kind, and extremely constructive. We will do our best to address all the limitations they pointed out (although we may not be able to fix everything within the conference deadline). We are also happy to hear that the idea was liked.

Most importantly, we want to express our sincere gratitude for the thorough and careful work you have done, something we do not take for granted, especially given past experiences with other conferences. We are convinced that only with the high quality reviews ( as the ones you did the science can advance and not so the conference is not simply a milestone to ad to the CV but an occasion to enhance the quality of our work).

We/I sincerely thank you.

---

### Note · Program_Chairs · 2026-01-17
**Submission Desk Rejected by Program Chairs**

The following references in this submission do not refer to real documents and/or have major errors in bibliographic information:

 Daniel Levy, Matthew D Hoffman, and Jascha Sohl-Dickstein. Normalizing flows for sampling, control, and variational objectives. arXiv preprint arXiv:1804.00779, 2021.
Alexander GdG Matthews, James Hensman, Richard Turner, and Zoubin Ghahramani. Gaussian process enhanced sampling for molecular dynamics. arXiv preprint arXiv:1703.10906, 2022.
Kim A Nicoli, Nils Strodthoff, Pan Kessel, Wojciech Samek, and Klaus-Robert Müller. Machine learning for statistical physics. Physical Review E, 104(4):046402, 2021.